# Molecular Genetics in Neuroblastoma Prognosis

**DOI:** 10.3390/children8060456

**Published:** 2021-05-29

**Authors:** Margherita Lerone, Marzia Ognibene, Annalisa Pezzolo, Giuseppe Martucciello, Federico Zara, Martina Morini, Katia Mazzocco

**Affiliations:** 1Unit of Medical Genetics, IRCCS Istituto Giannina Gaslini, 16147 Genova, Italy; margheritalerone@gaslini.org (M.L.); marziaognibene@gaslini.org (M.O.); federicozara@gaslini.org (F.Z.); 2IRCCS Istituto Giannina Gaslini, 16147 Genova, Italy; annalisapezzolo@gaslini.org; 3Department of Pediatric Surgery, IRCCS Istituto Giannina Gaslini, 16147 Genova, Italy; martucciello@yahoo.com; 4Dipartimento di Neuroscienze, Riabilitazione, Oftalmologia, Genetica e Scienze Materno-Infantili, University of Genova, 16132 Genova, Italy; 5Laboratory of Molecular Biology, IRCCS Istituto Giannina Gaslini, 16147 Genova, Italy; 6Department of Pathology, IRCCS Istituto Giannina Gaslini, 16147 Genova, Italy; katiamazzocco@gaslini.org

**Keywords:** neuroblastoma, genetics, TRK, liquid biopsy, exosomes, telomere maintenance, hypoxia

## Abstract

In recent years, much research has been carried out to identify the biological and genetic characteristics of the neuroblastoma (NB) tumor in order to precisely define the prognostic subgroups for improving treatment stratification. This review will describe the major genetic features and the recent scientific advances, focusing on their impact on diagnosis, prognosis, and therapeutic solutions in NB clinical management.

## 1. Introduction

Neuroblastoma (NB) is a pediatric heterogeneous disease with a median age of 17 months at diagnosis, which can evolve with a benign course or fatal illness, with a natural history ranging from a benign course to a terminal illness [1,2,3].

NB derives from neural crest progenitor cells with an overall incidence of 1 case per 100,000 children [4]. NB can arise in adrenal glands or in sympathetic ganglia with metastatic sites in bone marrow, lymph nodes, bone, liver, and orbital sites [5,6]. A subset of NB can spontaneously regress without treatment, while NB with widespread metastasis denotes a refractory disease and grim outcome [5]. The prognosis of NB ranges from spontaneous regression to progression, metastasis, and death with 5-year overall survival of more than 90% in the low-risk group, and about 40–50% in high-risk (HR) group patients [7,8]. Although this disease represents 8% of all malignant childhood cancers, it is responsible for 15% of pediatric cancer-related deaths [9].

NB patients have been classified into four categories (very low-, low-, intermediate-, high-risk) based on the presence of seven biological and clinical factors as suggested by the International Risk Group (INRG) [10]. The risk of death for each NB patient is defined based on disease presentation, age at diagnosis, tumor histology, tumor ploidy, localized or metastatic disease, recurrent segmental chromosome copy number alterations, and amplification of the proto-oncogene *MYCN* [7,10]. The current multimodal treatment of HR-NB includes surgery, chemotherapy, radiation, myeloablative chemotherapy with stem cell rescue, biological targeting, and immunotherapy [11,12]. The overall goals are minimizing surgical complications and reducing chemotherapy toxicity.

Tumor genetic analysis is a key component for risk stratification and prognosis in NB [7]. The molecular classification of NB tumors is currently routinely performed because of the influence of genetic variations both on treatment and clinical outcome. Recently, Coronado et al. suggested that the hemizygous deletion of chromosome 11q represents a biomarker of response to therapy with the anti-GD2 antibody combined with immune checkpoint inhibitors in HR-NB [13].

Although most cases of NB are sporadic and patients do not have a genetically inherited predisposition, rare familial cases present a genetic etiology in an autosomal dominant manner. [1]

In addition to the abovementioned biological and clinical factors, many genetic features contribute to better define NB patient prognosis. Past scientific efforts and the latest advances, supported by the employment of new technologies, contributed to the definition of the current NB risk stratification upon which the therapeutic treatment applied depends. In the last few years, the association between specific genetic variants and the risk of NB occurrence was discovered [14]. Genomic association studies (GWAS) identified risk variants in the following genes: *CASC15*, *BARD1*, *CHEK2*, *LMO1*, *LIN28B*, *AXIN2*, *BRCA1*, *TP53*, *SMARCA4*, and *CDK1NB* [1,14,15,16,17,18,19,20].

The purpose of the present review is to report the genetic implications known to date in the pathogenesis of NB and their correlation with the prognosis of the disease.

## 2. First Evidences of Genetics in NB

The patient’s age, clinical, and tumor classifications are the main factors affecting NB prognosis. Besides these, biological molecular markers have been introduced to better define the risk groups and to predict prognosis and disease recurrence.

### 2.1. MYCN

Among the *MYC* family of cellular proto-oncogenes, which regulates the expression of specific genes during growth and differentiation, *MYCN* and *c-MYC* are deeply involved in NB, the former evidently associated with a poor outcome [5]. Immunohistochemical studies demonstrated that tumors with normal levels of *MYCN* showed high levels of c-MYC, suggesting a correlation with poor outcome as well [21]. *MYCN* amplification, characterizing 25–30% of primary NB tumors, is the strongest independent prognostic factor for NB, regardless age and clinical stage, and it was the first used in risk classification [10]. It is highly associated with advanced stages of disease, rapid progression, and a poor prognosis, mainly found in HR-NB patients, but even in infants and patients with a lower stage of the disease [22,23]. *MYCN* was identified in 1983, and its genomic locus is the distal short arm of chromosome 2 (2p24.3) [21]. It is amplified both in primary tumors and cell lines, and cytogenetically identified in extrachromosomal acentric double minutes chromosomes and in homogenously staining regions other than chromosome 2. Dual-color FISH probes containing *MYCN* and *LAF* (2q11) control probes were used to assess *MYCN* status (amplified, not amplified) as recommended by the International Neuroblastoma Risk Group Biology Committee [24]. When the gene is amplified, it displays even more than 100 copies per nucleus. The amplified genomic region containing *MYCN* is usually on the order of 500 to 1000 kb, and additional genes located on 2p24.3 are frequently coamplified, such as *DDX1* in approximately 40–50% of NB and *NAG* [25,26,27] and *ALK* genes in about 10% of NBs. Moreover, *ALK* has also been validated as a direct target of *MYCN*-mediated transcriptional regulation [28].

### 2.2. Segmental Chromosome Alterations

Different conditions of ploidy have been described in NB, with whole-chromosome copy number variations usually associated with patients younger than 1 year of age and with an excellent outcome [29]. Finally, different segmental chromosomal alterations (SCA) with heavy prognostic impact have been identified and include loss of chromosomes 1p [30], 3p [31], 4p [30], 6q [32,33], and 11q [34], thought to express tumor suppressor genes, and gains of chromosomes 1q [35], 2p [36], and 17q [37], carrying supposed oncogenes. Gain of 17q has been reported in more than 50% of cases of neuroblastoma [37], and loss of 1p has been identified in one-third of cases [35]. Both gain of 17q and loss of 1p have been associated with *MYCN* amplification and a very poor prognostic outcome. On the contrary, loss of 11q, detectable in one-third of HR-NB cases, mainly in older patients, is inversely correlated with *MYCN* amplification but, nevertheless, is associated with a poor prognosis, too [35]. Indeed, NB tumors with a genomic profile characterized by any of these recurrent segmental alterations (SCA profile) are significantly associated with a high risk of relapse and a frequently poor outcome, especially when compared with NB tumors bearing only numerical alterations (NCA profile) [38]. NB patients bearing tumors with NCA have a favorable prognosis; nevertheless, a small percentage (10–15%) relapse locally or die. Recently, it has been reported that the loss of the whole chromosome X has a negative prognostic value for NB patients with an NCA genomic profile, and it was associated with a higher risk of relapse [39]. Patients affected by NB with a genomic profile displaying both SCA and NCA share the poor outcome of those with SCA only [8]. Other segmental aberrations (such as loss of 8p, 9p, 14q, 19p, 19q, and 22q, or gain of 5p, 12q, and others) are presently called atypical or not recurrent SCAs, and the risk of poor clinical outcome associated with these chromosomal alterations is not yet as well established as it is for the recurrent ones [40]. An SCA profile is associated with a higher risk of metastatic relapse among patients with localized disease, too, and among infants with localized unresectable or metastatic NB without *MYCN* amplification [41]. The accumulation of segmental chromosome alterations leads to NB progression, with particular regard to older children [42]. The high frequency of SCA-related to rare recurrent mutations in known protein-coding genes indicates that SCAs are driver events for this disease [40], and they have been incorporated into the patient risk stratification [38]. Since chromosomal breakpoints occur on several different loci, they probably reveal defects in DNA maintenance or repair mechanisms [42]; otherwise, an elevated amount of double-stranded DNA could break as a result of the specific cellular physiology of NB [43]. Recently, it has been shown that the distal 6q27 deletion characterizes a group of HR-NB patients with a particularly poor prognosis [32,33], suggesting that such an aberration can determine a negative clinical outcome due to the loss of function of the 14 genes mapping in such a region. The minimal 6q27 deletion contains three genes, *SFT2D1*, *RPS6KA2*, and *FGFR1OP*, that can promote an aggressive NB phenotype. *SFT2D1* is involved in vesicles transport, *RPS6KA2* has tumor suppressive functions in solid tumors, and the *FGFR1OP* gene encodes a centrosome protein for microtubules anchoring, and it is involved in both the proliferation and differentiation of erythroid lineage [44]. *FGFR1OP* haploinsufficiency can lead to abnormal erythropoiesis, and a reduced number of red blood cells has been observed in patients with NB [33]. Further studies are warranted to confirm the biological role of these genes in NB tumorigenesis.

### 2.3. PHOX2B and ALK

Familial NB is a quite rare event observed in about 1% of patients [45]. Linkage studies firstly identified missense or frameshift mutations in the paired-like homeobox 2B (*PHOX2B*) gene in a small group of familial NBs associated with Hirschsprung disease, congenital hypoventilation syndrome, and neurofibromatosis. *PHOX2B* is located at 4p13 and encodes a transcription factor essential in neurogenesis regulation. Germline mutations of *PHOX2B* occur in 6% of hereditary cases of NB [8,46,47], while in sporadic cases, they have been rarely observed [48].

Mutations in the anaplastic lymphoma kinase (*ALK*) gene, located in the 2p23 region, are the major factors leading to NB familial predisposition [49,50]. *ALK* is a receptor tyrosine kinase acting in neuronal differentiation [51], and it shows a high expression in the embryonic nervous system but will significantly decrease after birth. In NB, the somatically acquired genomic amplification and activating mutations of *ALK* occur in 2–3% and 8–10% of primary tumors, respectively [48,52,53], and play an important role in NB oncogenesis [54], evidently associated with a poor outcome [4,5]. Germline mutations in the tyrosine kinase domain of *ALK* are different from that of the somatic mutations in sporadic cases: of the three main hotspot mutations, only the R1275 mutation has been recurrently observed in familial cases [48,55]. With regard to the F1174 and F1245 mutations, they have not been reported in familial cases, suggesting a more aggressive behavior with respect to the R1275 one [56]. Mutations that cause a constitutive activation of *ALK* result in an oncogenic activity, affecting downstream signaling pathways, such as the *RAS/MAPK* and the *RET* ones, and inducing cell transformation [57].

### 2.4. TRK

Somatic chromosomal translocations involving the *NTRK1*, *NTRK2*, and *NTRK3* genes occur in approximately 1% of tumors. The tropomyosin-related kinase (Trk) family includes three receptor tyrosine kinases (RTKs) named TrkA, TrkB, and TrkC that are regulated by neurotrophins, growth factors involved in neuronal development and function. Each receptor shows specific affinity for different ligands: TrkA binds NGF (nerve growth factor), TrkB recognizes BDNF (brain derived neurotrophic factor), and TrkC binds NT-3 (neurotrophin-3) [58,59].

The activation of the different receptors can lead to distinct cellular outcomes, including neuronal differentiation, cell survival, development, and synaptic plasticity [60,61,62]. The neurotrophins–Trks binding activates the downstream Ras/MAPK and PI3K/Akt-mTOR signaling pathways, central for cell proliferation and survival. Activation of TRK family receptors is implicated in both pediatric and adult tumors. Indeed, many childhood tumors are characterized by *TRK* fusions. The specific *EVT6–NTRK3* fusion has been detected in more than 75% of cases of certain rare cancers such as cellular congenital mesoblastic nephroma, infantile fibrosarcoma, and secretory carcinoma of the breast [63,64,65]. In other pediatric cancers, such as in spitzoid melanomas and in high grade glioma, the *TRK* fusion incidence is lower (from 10% to 40%). NBs that spontaneously regress or show high differentiation levels are characterized by high expression of TrkA, suggesting its favorable prognostic impact, while in the majority of HR-NB NBs, the overexpression of TrkB is associated with an unfavorable outcome and aggressive tumor progression. BDNF/TrkB signaling promotes cell survival and angiogenesis. Indeed, inhibiting TrkB activity leads to cell apoptosis and increased chemosensitivity [66,67]. On the contrary, TrkC is overexpressed in favorable neuroblastomas. Given the significant oncogenic role of TRK fusion proteins, their inhibition could provide an effective therapeutic strategy for NB. Recently, clinical trials have reported the use of larotrectinib, a highly selective pan-TRK inhibitor, in children with a change in *NTRK1*, *NTRK2*, or *NTRK3* genes, which demonstrated a good response.

### 2.5. Mutations in Genes Other Than MYCN with Prognostic Value in NB

Despite MYCN oncogene aberrations playing a major role in NB development, many studies highlighted the presence of genetic variations in other genes contributing to NB occurrence. *LIN28B* exerts a negative regulation of let-7 miRNAs, leading to increased MYCN protein levels, and its overexpression or amplification in HR-NB tumors has been associated with a poor prognostic outcome [68]. *LIN28B* can also promote RAS-related nuclear protein (RAN) expression through direct binding of RAN mRNA or by downregulating the Ran stabilizing protein RANBP2 through let-7 miRNA suppression [69]. Moreover, *LIN28B* contributes to metastases formation by increasing the invasive and migratory ability of tumor cells and, thus, confers the aggressive phenotype HR-NB tumors [70,71]. Consequently, targeting *LIN28B* provides a valuable therapeutic approach. Difluoromethylornithine (DFMO) inhibits ornithine decarboxylase (ODC), which indirectly affects the LIN28B-mediated biological axis. DFMO treatment induced *LIN28B* downregulation, inhibiting NB tumor progression by decreasing the glycolytic metabolic rate of NB tumor cells [72]. Moreover, BET bromodomain inhibitor JQ1, in combination with panobinostat, showed strong antitumor effects in NB: it blocked BRD3 and BRD4 activity, preventing *LIN28B* transcriptional activation [73].

A GWAS study identified specific variants of *BARD1* and *LMO1* associated with HR-NB disease [74]. Interestingly, while the full length *BARD1* transcript encodes for a protein with tumor suppressive function, alternative splicing variants produce protein products with opposite oncogenic properties [75]. Metastatic NB cases show a higher expression of the *BARD1*β transcript, which enhances tumor growth and leads to the acquisition of the most aggressive traits of the disease [76].

*LMO1* exerts an oncogenic role: genetic variants occurring in this gene were associated with higher susceptibility to developing aggressive NB tumors, and its overexpression, occurring through chromosomal duplication events, was associated with a poor prognostic outcome [77].

The majority of adult tumors are characterized by mutations in *TP53* gene encoding for the p53 protein, which plays a pivotal role in cell cycle regulation. Thus, aberrations in p53 function promote tumor development [78]. On the contrary, pediatric tumors, including NB, do not display *TP53* mutations, but p53 activity is altered by the aberrant expression of *MDM2*, which negatively regulates p53 [79]. It has been demonstrated that higher expression levels of *MDM2* in *MYCN* nonamplified NB patients have a prognostically negative impact [80]. Recently, two genomic amplifications on chromosome 12 have been observed, involving the mapping sites of *MDM2*, *CDK4*, and *FRS2* genes, and these aberrations were correlated with NB patients’ poor clinical outcome [81]. In vitro studies on *MYCN*-amplified NB cell lines showed that *MDM2* is able to act independently from p53 and to interact with the *MYCN* transcript, increasing its stability and, thus, favoring its expression [82]. Therefore, the protumorigenic effects of *MDM2* occur both in *MYCN* nonamplified tumors via the p53 pathway and in *MYCN*-amplified tumors in a p53-independent mechanism [83]. Considering its strong tumor-enhancing properties, *MDM2* has become an attractive potential therapeutic target. Nutlin-3 was the first developed small molecule effective in antagonizing MDM2 activity and counteracting tumor growth [84]. Further studies led to the production of even more effective and specific second-generation *MDM2* inhibitors. Among them, RG7388 was able to significantly reduce tumor burden in NB xenograft models [85] and, when combined with chemotherapeutic drugs commonly employed for NB treatment, it showed synergistic effects in inducing tumor cell apoptosis [86].

## 3. Brand New Emerging Genetic Implications in NB

Translational research in clinical oncology has been highly affected by the postgenome era, which has led to the application of new technologies including next generation sequencing, gene expression, and proteomics analysis. These methods allow us to obtain large data sets to better understand the molecular profile of tumors, with the possibility of characterizing tumor heterogeneity and defining targeted therapeutic approaches. Beside the development of novel technologies, translational research has seen an increasing interest in the study of liquid biopsies for monitoring the evolution of the disease and patient response to treatment. Both aspects characterized recent relevant studies on NB.

### 3.1. Exploring the Prognostic Value of Liquid Biopsy-Derived Markers in NB

Primary tumor tissue analysis is the golden standard for the molecular characterization of oncologic diseases. However, tissue sampling is highly limited by the invasiveness of the surgical procedure and by the quality of biological material that is not always adequate for biomolecular analyses. The need for novel analytical tools in clinical oncology has led to the investigation of liquid biopsies as a compelling, less invasive alternative for tumor analysis. In particular, HR-NB tumors often infiltrate adjacent structures [87] interfering with tissue sampling and, thus, making liquid biopsies an essential tool for studying tumor biology. Biological fluids are a valuable source of circulating messenger RNA (mRNA), tumor DNA (ctDNA), cell-free DNA (cfDNA), and circulating tumor cells (CTCs) that may provide biomarkers for tumor diagnosis and prognosis [88].

#### 3.1.1. Liquid Biopsies to Unveil Minimal Residual Disease (MRD)

The concept of MRD refers to the persistence of cancerous cells after the conclusion of the chemotherapeutic treatment. These drug-resistant tumor cells are the main cause of relapse occurrence in NB patients, often resulting in fatal clinical outcome [5]. Therefore, the accurate detection of MRD is critical to improve NB patient prognosis. Residual NB tumor cells responsible for MRD can be often detected in peripheral blood (PB) and bone marrow (BM) [89], making liquid biopsies the optimal method for such analysis.

The first efforts in investigating blood samples of NB patients showed that high levels of circulating tyrosine hydroxylase (*TH*), *PHOX2B*, and doublecortin (*DCX*) mRNAs in peripheral blood (PB) and bone marrow (BM) samples at diagnosis are indicative of poor treatment response and worse clinical outcome [90]. These results were obtained through real-time quantitative PCR (RQ-PCR) analysis, which is the most sensitive method for MRD detection. Stutterheim et al. showed that the evaluation of two different panels of RQ-PCR markers can increase the sensitivity of MRD assessment. In particular, authors identified a panel for BM samples analysis (including *PHOX2B*, *TH*, *DDC*, *CHRNA3*, and *GAP43*) and a panel for PB samples (including *PHOX2B*, *TH*, *DDC*, *DBH*, and *CHRNA3*). The assessment of both RQ-PCR marker panels is crucial to obtain the highest sensitivity for a proper MRD detection [91].

A recent Japanese study demonstrated the high prognostic value of a 7 NB-mRNA signature (*CRMP1*, *DBH*, *DDC*, *GAP43*, *ISL1*, *PHOX2B*, and *TH*) evaluated with droplet digital PCR (ddPCR), a technique providing higher accuracy and reproducibility than RQ-PCR [92]. It was shown that ddPCR outperforms RQ-PCR in detecting the expression of the 7 NB-mRNA signature in BM and PB samples of HR-NB patients.

Besides mRNA signature analysis, the residual circulating tumor cells can be isolated from blood samples. It has been demonstrated that CTCs are present in BM samples of HR-NB patients both at diagnosis and at relapse. In particular, CTC genomic analysis provided comparable results to primary tumor tissue analysis, allowing for the proper detection of the aberrations occurring in genome coding regions [93].

These results demonstrate that liquid biopsies may be instrumental for NB diagnosis when primary tissue sampling is not feasible.

#### 3.1.2. CfDNA and CtDNA Analyses for Understanding NB Clonal Evolution and Relapse Occurrence

The amount of cfDNA has been correlated with tumor burden in NB patients [94], and it provides important information for understanding tumor heterogeneity, a main hallmark of NB.

CtDNA analysis may serve as a surrogate for tissue biopsy to evaluate the genomic profile of NB tumors, and several studies have focused on the feasibility of detecting the genetic variants of the primary tumor in circulating nucleic acids. Successful results were obtained for the assessment of both *MYCN* [95] and *ALK* status by analyzing ctDNA. The sensitivity and specificity of the results were ensured by the application of droplet digital PCR technology (ddPCR), which confirmed the feasibility of determining the *MYCN* and *ALK* copy number profile from blood samples [96,97]. As the presence of the segmental chromosomal alterations strongly affects NB prognosis, the possibility to assess the NB genomic profile by analyzing cfDNA when tumor DNA is not available has been investigated. It has been shown that cfDNA in serum and plasma samples can be used to detect a 17q gain in NB patients [98]. Similarly, it has been demonstrated that shallow whole genome sequencing on cfDNA can identify the same chromosomal aberrations detected in primary tumor tissue and, thus, cfDNA can be considered as a reliable alternative source for the study of NB molecular features [99].

Recent studies showed that cfDNA analysis allows us to detect different cellular subclones at diagnosis, each carrying specific genomic aberrations, which can be responsible for treatment resistance or relapse occurrence [100]. Studying the spatial and temporal evolution of such clones through cfDNA analysis represents an important tool for disease monitoring and for developing targeted treatment strategies. Nevertheless, the low burden of recurrent mutations in NB [101] hinders the detection of NB specifically derived cfDNA and, thus, to overcome such a limitation, novel deep sequencing technologies are required. To this purpose, recent studies focused on the evaluation of the methylation status of cfDNA. Applebaum M. et al. demonstrated that the level of 5-hydroxymethylcytosine, a marker of active gene expression, is associated with disease burden and is indicative of treatment response and relapse occurrence [102]. Van Zogchel LMJ et al. [103] proposed the analysis of hypermethylation of *RASSF1A* (*RASSF1Am*) as a circulating biomarker for ctDNA detection, showing that *RASSF1Am* combined with the mRNA signature of BM analysis is able to better stratify patients with minimal residual disease [103].

The efforts of optimizing ctDNA analysis are leading to new insights in NB diagnosis and prognosis, but the stability of ctDNA and cfDNA is threatened by nucleases that can easily degrade circulating-free molecules: the short half-life of cfDNA can provide a “real-time” picture of disease status, but several different factors affecting the clearance rate should be considered [104].

#### 3.1.3. Circulating Exosomes for NB Patient Response and Chemotherapy Resistance

Biological fluids also contain circulating exosomes, small vesicles involved in cellular communication, which provide a more stable intravesicular environment. The content of the exosomes probably depends on what stress the cell is experiencing, and it can change over time depending on the different stressors or stimuli to which the cell is subjected [105]. The discovery that the contents of exosomes can be transmitted from one cell to another by fusion supports the idea that exosomes are dynamic mediators of cellular communication [106]. All these studies have led to defining that exosomes are circulating small vesicles containing nucleic acids, miRNA, and proteins that are able to strongly affect cell behavior. The exosomes are released by most cell types, including tumor cells. Indeed, higher amounts of exosomes are secreted in pathological conditions. Exosomes cargo reflects the content of the tumor cells of origin, representing a valuable source of cancer biomarkers [107,108,109,110,111].

It has been demonstrated that exosomal-microRNA (exo-miR) derived from the plasma of HR-NB patients is predictive of treatment response [112]. In particular, exo-miR-29c, -342-3p, and let-7b downregulation correlated with a poor response to induction chemotherapy treatment. Moreover, the exo-miR expression profile is able to provide a chemoresistance index predicting sensitivity/resistance to specific chemotherapeutic drugs, allowing us to define targeted treatment approaches for maximal HR-NB patient response [112].

In vitro studies confirmed the role of exo-miR in chemoresistance, as reported by Challagundla et al. [113], who demonstrated the importance of the exosome mediated cross talk between NB cells and the tumor microenvironment (TME). In particular, it was shown that NB cells transfer miR-21 to monocytes via exosomes. MiR-21 leads to the upregulation of monocyte miR-155 expression in a Toll-like receptor-8-(TLR8)-dependent manner. MiR-155 is then uploaded in monocyte-derived exosomes and delivered back to NB cells, where it induces the downregulation of TERF1, a telomerase inhibitor, leading to increased telomerase activity and, thus, chemotherapy resistance.

Considering all this evidence highlighting the potential prognostic value of liquid biopsies in NB, the latest research studies [114] aim at optimizing the standard operating procedures (SOPs) for the study of circulating molecules and exosomes. The establishment of SOPs and the prospective validation of the results will be mandatory for the application of liquid biopsies assessment in a clinical setting for NB diagnosis, treatment, and monitoring.

#### 3.1.4. Exosomal Double-Stranded DNA for the Analysis of Tumor Mutational Profile

Exosomes released by cancer cells contain about 20 times more exosomal-DNA (exo-DNA) than normal fibroblast-derived exosomes [107]. The exosomes isolated from tumor cells included exo-DNA that can be longer than 10 Kb [108]. It has recently been shown that the exo-DNA contained in the exosomes of NB patients can be useful for the analysis of the parental tumor mutational profile [115]. Degli Esposti et al. demonstrated, by whole exome sequencing, that NB-plasma-derived exosomes contain >10 Kb fragments of double-stranded tumor DNA. The exo-DNA showed genetic mutations in known NB oncogenes and tumor suppressor genes that were detected also in the tumor of origin. The most frequently analyzed somatic mutations in NB exo-DNA occurred in *ALK*, *CHD5*, *SHANK2*, *PHOX2B*, *TERT*, *FGFR1*, and *BRAF* genes. Furthermore, high genomic amplification of *MYCN*, *TERT*, and *SHANK2* genes has been observed [115]. Exo-DNA of NB relapsed patients carried mutations in *ALK*, *TP53*, and *RAS/MAP* genes, suggesting that these somatic genetic variants may be responsible for acquired treatment resistance [115]. Notably, a considerably higher number of somatic mutations has been identified in exo-DNA at relapse than at diagnosis, supporting the theory of a clonal evolution. Certainly, circulating exosomes in the plasma of NB patients can serve as a source for the analysis of somatic mutations occurring in the primary tumor. Degli Esposti et al. [115] provided evidence for the analysis of a novel source of circulating DNA that represents an alternative to cfDNA, which can derive from cell lysis or apoptosis [96,99,101]. Therefore, the exo-DNA released by living cells better depicts the tumor dynamics and the most aggressive cell subpopulations. More specifically, the exo-DNA is stable and less fragmented than cfDNA because it is protected within an intravesicular environment and, thus, it represents a more reliable biological source to be analyzed in a clinical setting. Interestingly, exo-DNA also captures metastases features that cannot be detected by only analyzing needle biopsies of the primary tumor. In fact, somatic mutations that were not present in tumor DNA at the onset were instead identified in exo-DNA, in particular variants of *ALK*, *ATRX*, *NF1*, and *TERT* genes [115]. In this regard, exo-DNA can be instrumental for analyzing the great intratumoral spatial heterogeneity typical of NB [100], and for developing targeted therapeutic approaches. In the future, exo-DNA could provide a noninvasive method for diagnostic purposes, risk classification and patient stratification, and monitoring the response to therapy for NB patients, especially in those cases in which tumor tissue biopsy would not be feasible nor informative.

### 3.2. Telomerase Activity in Neuroblastoma

*MYCN* amplification is considered essential to evaluate the disease and to stratify treatment, but this is not sufficient to ensure an accurate prognostic risk grouping [116,117].

The identification of novel prognostic markers can improve not only the accuracy of risk assessment but also the definition of targets to develop new therapies. An emerging new marker is telomerase which, by its action on telomere maintenance, significantly contributes to cell immortalization, drug resistance, and tumorigenesis in almost 90% of human tumors, including peripheral neuroblastic tumors [118].

Telomeres are repetitive nucleotide sequences rich in guanine residues at the ends of chromosomes with the main function of maintaining genomic integrity. It is known that telomere dysfunction, mainly due to telomere capping disruption, promotes breakage-fusion-bridge (BFB) cycles, resulting in chromosome instability (CIN) [118]. Such events contribute to the structural and numerical aneuploidy observed in the majority of tumors, including NB [119]. It has been hypothesized that aneuploidy and CIN induce intratumor heterogeneity, a hallmark exploited by cancer cells to increase their adaptive potential and, thus, their survival [120,121]. It is also known that the absence of a replication system able to properly preserve telomere length in somatic cells leads to the progressive physiological shortening of telomeres, which is responsible for replicative senescence and apoptosis. To escape this critical condition, cancer cells promote the re-expression of telomerase reverse transcriptase (*TERT*), the catalytic subunit of telomerase [122,123]. Thus, telomere dysfunction is an impermanent process required for CIN-promoted tumor initiation, but telomerase reactivation is subsequently needed to ensure tumor progression [124].

To maintain telomeres length, tumor cells are also able to activate an alternative lengthening of telomeres (ALT) mechanism, based on homologous recombination. In particular, tumors developing from mesenchymal tissues are mainly characterized by ALT activity [125]. NB tumors are included in such a category, as they originate from the peripheral nervous system. It has been demonstrated that the coexistence of cancer cell subpopulations with different telomere length within NB is significantly associated with poor clinical outcome and disease progression in NB patients [126]. The study by Pezzolo et al. suggested that the coexpression of ALT mechanism and *TERT*, observed in 60% of the NB tumors analyzed, may play a major role in NB tumor progression. Importantly, telomere maintenance mechanisms are associated with poor clinical outcome regardless of clinical stage and their activation has been observed in HR-NB tumors but not in localized NB cases. It is known that *TERT* is a direct target of the *MYCN* oncogene and 40% of *MYCN*-amplified HR-NB cases are associated with *MYCN*-induced up-regulation of *TERT* [127]. Besides *MYCN* regulation, *TERT* overexpression can occur through other mechanisms, including gene amplification, genomic rearrangements, and mutations within the promoter region. The absence of *TERT* promoter mutations in NB tumor samples allows us to exclude this mechanism as the main one responsible for *TERT* overexpression in NB [128,129].

In the whole-genome sequencing analysis of neuroblastoma, recurrent genomic rearrangements of the catalytic subunit of *TERT* (5p15.33) and of the alpha thalassemia/mental retardation syndrome X-linked (*ATRX*) genes have been identified [127], in addition to *MYCN* amplifications, *ALK*, and *PHOX2B*. A subgroup of HR-NB tumors is characterized by the active ALT mechanism, whose function is negatively regulated by the *ATRX* gene [130]. Indeed, *ATRX* missense, nonsense, and frameshift mutations have been identified in NB tumors and associated with ALT activation and poor outcome in *MYCN* nonamplified NB tumors [131,132]

Moreover, *ATRX* deletions have been reported in 11% of HR NBs [133,134] and result in the loss of the nuclear *ATRX* protein and ALT mechanism activation. Recent evidence revealed that patients with *TERT* or ALT activation and harboring alterations in the RAS/p53 cellular pathway are identified as very HR cases, who are prone to relapse and associated with a very poor clinical outcome [135].

Telomerase maintenance is considered a powerful prognostic marker for NB patients and, thus, it provides a target for potential novel therapeutic treatments. A recent study tested the efficacy of the BET bromodomain inhibitor OTX015, targeting the BET bromodomain protein BRD4 that induces *TERT* expression, and carfilzomib, a proteasome inhibitor [136]. The results showed a strong synergistic effect of the two drugs in blocking *TERT* overexpression, inducing NB cell apoptosis in vitro, and drastically reducing tumor progression in murine models. These findings can encourage the development of the first clinical trial based on a combination of OTX015 and carfilzomib for patients harboring *TERT*-rearranged NB tumors.

### 3.3. Hypoxia

Hypoxia is a main hallmark of solid tumors, and it is represented by a decreased concentration of oxygen within tumor masses. This condition has been associated with aggressive cancer phenotypes characterized by higher metastatic potential and chemoresistance [137,138]. A pioneer study investigating the relationship between hypoxia and neuroblastoma has been published by Fardin P. et al., who first determined a hypoxia gene signature (NB-hypo) by analyzing the gene expression of different NB cell lines [139].

Gene expression profiling performed on primary NB tumors revealed that the identified NB-hypo could efficiently differentiate NB patients with good and poor clinical outcomes, showing for the first time that hypoxia is an independent prognostic factor for NB risk stratification [140]. Further studies refined the NB-hypo by reducing the hypoxia-associated gene expression signature to seven genes (NB-hop). The signature pointed out that the overexpression of the *FAM162a, PDK1, PGK1*, and *MTFP1* genes and the downregulation of the *ALDOC* gene were associated with an unfavorable outcome for NB patients [141]. Importantly, the NB-hop was able to further stratify unfavorable subgroups of patients among low-risk cases without *MYCN* amplification and intermediate-risk patients with a stage 3, *MYCN* nonamplified disease who were older than 18 months [141]. These results were obtained in the most numerous cohort of NB patients analyzed, thanks to the application of a novel complex bioinformatics pipeline able to make the expression data of different datasets comparable [141]. Thus, hypoxia was confirmed as an independent prognostic factor predicting NB patient clinical outcome. Furthermore, the unfavorable NB-hop expression was strongly correlated with telomerase activation and immunosuppressive tumor microenvironment [141].

Considering all these findings, hypoxia represents a valuable therapeutic target in NB. Hypoxia-activated prodrugs (HAPs) are specifically designed to target the hypoxic areas of tumors, as these compounds are activated via oxidoreductase-mediated reduction, which is irreversible in the absence of oxygen [142]. Recent evidence showed that the administration of Evofosfamide, a drug belonging to the HAP class, produced cytotoxic effects both in vitro and in NB xenograft models, where a reduction in tumor growth was reported. The antitumor activity of Evofosfamide was amplified when used in combination with Topotecan [143].

The high specificity of HAPs in impairing hypoxic tumor cell viability can exert an important therapeutic activity in cancer patients who are properly stratified according to the hypoxic tumor status [142]. Thus, scientific efforts aiming at identifying hypoxic signatures able to properly stratify NB patients are mandatory for assessing the efficacy of HAPs.

## 4. Conclusions

In recent years, many advances have been made in NB genomics. Numerous studies have been published on recurrent somatic mutations and chromosomal abnormalities that contribute significantly to NB genesis, but their role in NB metastasis and response to therapy has not been fully understood yet. The broad range of phenotypical heterogeneity in NB, partly due to the diverse genetic features of this tumor, makes the identification of druggable genetic traits mandatory. Recurrent chromosomal rearrangements predict the prognosis of NB, but the genes mapping in such loci and involved in NB tumorigenesis have still to be completely defined. The present review summarized the main genetic features affecting NB prognosis studied in both primary tumor tissue and liquid biopsies (Table 1).

The advent of the genomic era has provided significant improvements in NB genetic characterization, and the possibility to analyze big datasets of patients has surely given essential results for better understanding the genetic features involved in NB tumorigenesis and progression. NB is a fatal pediatric tumor for 50% of HR-NB patients, whose cure largely depends on the possibility of studying the expression of biological markers of tumor progression and response to therapy. Thus, biological tumor profiling is mandatory and it is usually performed on primary tumor tissue. NB tumor samples have been essential to characterize many molecular features of NB tumors; however, a high percentage of NB patients, especially the high-risk class, are characterized either by tissue biopsies with insufficient content of neoplastic cells or by the absence of tumor tissue to be analyzed. Needle biopsy analysis is limited (i) by the difficulty of sampling an adequate amount of tissue, (ii) by the invasive nature of this method that hinders its repeatability, and (iii) by the impossibility to capture tumor heterogeneity. All these reasons justify the strong interest of the scientific community in alternative and more easily accessible biological material, such as the liquid biopsy. Recent scientific efforts in the study of liquid biopsies have revealed the importance of biological fluids for monitoring treatment response and for the identification of specific mutations in NB cellular subclones, undetectable in the primary tumor and responsible for relapse occurrence. Such mutations can serve as novel therapeutic targets for the development of targeted therapeutic treatment aiming at improving the survival of HR-NB patients.

## Figures and Tables

**Table 1 children-08-00456-t001:** Markers of unfavorable prognosis in NB.

Biomarker	Biological Source	Genetic Event	Alteration Result	References
*MYCN*	Primary tumor/cfDNA/exoDNA	Amplification	Overexpression	[5,21,22,23,95,96]
*PHOX2B*	Primary tumor/PB and BM/exoDNA	Somatic mutations	Overexpression	[46,47,90,91,92]
*ALK*	Primary tumor/cfDNA/exoDNA	Amplification and somatic mutations	Overexpression	[48,49,50,52,53,54,55,56,57,97,115]
*TRK*	Primary tumor	Fusion	Overexpression	[66,67]
*LIN28B*	Primary tumor	*MYCN*-mediated transcriptional activation	Overexpression	[68,69]
*BARD1*	Primary tumor	Alternative splicing	Acquired oncogenic properties	[74,75,76]
*LMO1*	Primary tumor	Duplication	Overexpression	[74,77]
*MDM2*	Primary tumor	Amplification	Overexpression	[79,80,81,82,83]
*CDK4*	Primary tumor	Amplification	Overexpression	[81]
*FRS2*	Primary tumor	Amplification	Overexpression	[81]
*TH*	PB and BM	-	Overexpression	[90,91,92]
*DCX*	PB and BM	-	Overexpression	[90]
*DDC*	PB and BM	-	Overexpression	[91,92]
*CHRNA3*	BM	-	Overexpression	[91]
*GAP43*	PB and BM	-	Overexpression	[92]
*DBH*	PB and BM	-	Overexpression	[91,92]
*CRMP1*	PB and BM	-	Overexpression	[92]
*ISL1*	PB and BM	-	Overexpression	[92]
*RASSF1A*	PB and BM	Hypermethylation	-	[103]
*CHD5*	exoDNA	Somatic mutations	-	[115]
*SHANK2*	exoDNA	Somatic mutations	-	[115]
*FGFR1*	exoDNA	Somatic mutations	-	[115]
*BRAF*	exoDNA	Somatic mutations	-	[115]
*TP53*	exoDNA	Somatic mutations	-	[115]
*NF1*	exoDNA	Somatic mutations	-	[115]
*TERT*	Primary tumor/exoDNA	Amplification and rearrangements	Overexpression	[118,127]
*ATRX*	Primary tumor/exoDNA	Rearrangements and somatic mutations	Overexpression	[127,130,131,132]
*FAM162a*	Primary tumor	-	Overexpression	[141]
*PDK1*	Primary tumor	-	Overexpression	[141]
*PGK1*	Primary tumor	-	Overexpression	[141]
*MTFP1*	Primary tumor	-	Overexpression	[141]
*ALDOC*	Primary tumor	-	Downregulation	[141]
miR-29c	Exosomes	-	Downregulation	[112]
Let-7b	Exosomes	-	Downregulation	[112]
miR-342	Exosomes	-	Downregulation	[112]

The table shows the list of molecular markers that define NB prognosis. For each marker, the biological source, the type of genetic alteration, and the subsequent aberrant expression associated with unfavorable NB prognosis have been reported. Abbreviations: PB = peripheral blood; BM = bone marrow; exoDNA = exosomal DNA.

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
