# Peer review of "Molecular Genetics in Neuroblastoma Prognosis"

_children, 2021, doi:10.3390/children8060456_

Round 1

Reviewer 1 Report

The present Review article reports about actual state of molecular genetics in neuroblastoma, focused on prognostic impact.

It is a very well written overview about molecular genetics in neuroblastoma.

The paper is well organized and profound review of actual literature is clearly visible.

The abstract is short and on the point clear and invites to read the whole paper.

Introduction is leading to the purpose of the review - reporting the genetic implications in the pathogenesis of neuroblastoma.

The main chapter, devided in»First evidences of genetics in NB» and «Brand new emerging genetic implications in NB» shows in a dynamic matter the recent development in research on this topic. It provides an excellent overview for interested physicians and researcher, in a broad understandable manner.

Conclusions are as clear as possible and next to the general aspect, as usually in reviews, there is an outview in direction of targeted therapies in the sense of translational research and results of scientific workup.

Reviewer 2 Report

This review is well-written but the similar reviews have already published. `In recent standpoint, telomere maintenance is suggested as a "driver " of unfavorable NB. The segmental chromosome aberration might be the result of chromosome instability due to telomere crisis. After telomere crisis, telomerase activation of ALT occur in unfavorable  NB. Therefore, the authors should be the correlation among these genetic aberrations.
